# Citizen Science Monitoring for Sustainable Development Goal Indicator 6.3.2 in England and Zambia

**Isabel J. Bishop** [1,]*[ID]**, Stuart Warner** [2][ID]**, Toos C. G. E. van Noordwijk** [1]**, Frank C. Nyoni** [3] **and Steven Loiselle** [1]

[1] Earthwatch Europe, Oxford OX2 7DE, UK; tvnoordwijk@earthwatch.org.uk (T.C.G.E.v.N.); sloiselle@earthwatch.org.uk (S.L.)

[2] United Nations Environment Programme GEMS/Water Capacity Development Centre, School of Biological, Earth & Environmental Sciences and Environmental Research Institute University College Cork, T23 XE10 Cork, Ireland; s.warner@ucc.ie

[3] Water Resources Management Authority (WARMA), Lusaka 10101, Zambia; fcnj8p@yahoo.com

* Correspondence: ibishop@earthwatch.org.uk

**Abstract:** Citizen science has the potential to support the delivery of the United Nations Sustainable Development Goals (SDGs) through its integration into national monitoring schemes. In this study, we explored the opportunities and biases of citizen science (CS) data when used either as a primary or secondary source for SDG 6.3.2 reporting. We used data from waterbodies with both CS and regulatory monitoring in England and Zambia to explore their biases and complementarity. A comparative analysis of regulatory and CS data provided key information on appropriate sampling frequency, site selection, and measurement parameters necessary for robust SDG reporting. The results showed elevated agreement for pass/fail ratios and indicator scores for English waterbodies (80%) and demonstrated that CS data improved for granularity and spatial coverage for SDG indicator scoring, even when extensive statutory monitoring programs were present. In Zambia, management authorities are actively using citizen science projects to increase spatial and temporal coverage for SDG reporting. Our results indicate that design considerations for SDG focused citizen science can address local needs and provide a more representative indicator of the state of a nation's freshwater ecosystems for international reporting requirements.

**Keywords:** sustainable development goals; citizen science; indicator 6.3.2 ambient water quality; freshwater watch

## 1. Introduction

Citizen science (i.e., the involvement of non-scientists in scientific research) has been increasingly recognized as having a potentially important role to play in delivering the United Nations' (UN) Sustainable Development Goals (SDGs). Central to this role is the ability of citizen science to provide data for SDG indicators, which are the mechanism through which the progress of UN member states measure SDGs [1]. The use of citizen science in this way presents several opportunities for improving the knowledge base that underpins global progress for SDGs, including increasing the frequency of available data, expanding the geographic reach of data, addressing gaps in funding, educating the public about relevant policy issues, and making better use of local knowledge [2].

In a systematic review of all 244 SDG indicators, Fraisl et al. [3] identified the targets and indicators to which citizen science could make the greatest contribution. Alongside SDG 3 (good health and wellbeing), SDG 11 (sustainable cities and communities), and SDG 15 (life on land), SDG 6

(clean water and sanitation) was identified as a goal that could greatly benefit from citizen science. In particular, citizen science is well placed to contribute to indicator 6.3.2 (proportion of bodies of water with good ambient water quality). Although an established methodology for 6.3.2 exists (UN Water 2018), data are not regularly produced by conventional methods because many member states, and particularly those in the least developed countries, lack the resources required to establish water quality monitoring programs at high spatial and temporal resolutions [4]. Compared with many other indicators, reporting on indicator 6.3.2 is inherently more difficult and currently relies on an established monitoring network and an institutional capacity to collect, manage, and assess water quality data. In Europe, more or less extensive regulatory water quality monitoring programs already exist for other policy purposes, such as the EU Water Framework Directive [5]. However, this prerequisite is beyond the capacity of many countries [6] and helps explain why progress towards target 6.3, as well as other Goal 6 targets, is behind schedule and not on track to be realized by 2030 [7].

Citizen science monitoring for water quality is well established, with volunteers in many parts of the world using low-cost kits and sensors to contribute spatially and temporally resolute data as part of a variety of different local, national, and sometimes international initiatives [8,9]. Fritz et al. [1] created a roadmap for citizen science and SDG reporting, including identification of relevant citizen science projects, ensuring data quality, and mobilizing and integrating stakeholders and communities in UN member states. Significant progress through this roadmap has already been made with regards to indicator 6.3.2, in particular by the citizen science project Freshwater Watch (FWW). Since 2012, over 10,000 citizen scientists from 29 countries across five continents have contributed ~25,000 measurements of ambient water quality to FWW. These measurements include two of the five core parameters for indicator 6.3.2 (phosphates and nitrates), as well as one supplementary ('level 2') parameter (turbidity). Data generated by citizen scientists using FWW has been specifically validated for SDG reporting by comparison to accredited laboratory methodologies [10]. Progress has also been made in mobilizing relevant policy stakeholders, for example FWW has been identified by the European Commission as a project suitable for policy use [11]. Several national agencies responsible for water quality monitoring in regions where data is scarcer, including the Water Resources Management Authority of Zambia, are now looking to incorporate FWW into their national reporting. For this to happen, one significant step in Fritz et al.'s roadmap must still be tackled: the integration of citizen science data streams into the practices of National Statistics Offices (NSO).

In this study, we examine the potential for FWW to contribute to indicator 6.3.2 reporting via integration into national scale water quality monitoring systems in two contrasting case study countries: (a) England (UK), where the regulatory monitoring infrastructure is well-developed with a large number of measuring sites determined for local and national requirements, and where FWW activity has developed organically through use in multiple locally-focused projects, and (b) Zambia, where regulatory monitoring infrastructure is expanding from a limited basis, and where there is a desire to expand existing FWW activity to a national scale. Specifically, we ask the following questions:

1.  Can existing citizen science (FWW) data be used to produce a reliable 6.3.2. indicator score?
2.  What value does integration of citizen science data add to indicator 6.3.2 reporting?
3.  How can the benefit of citizen science schemes for the purpose of SDG reporting be maximized?

To answer these questions, we compare the datasets generated by both monitoring schemes (regulatory and FWW) in each of the case-studies to uncover the biases and complementarities of the schemes. For England, where a substantial number of water bodies was monitored by both schemes, we also directly compare indicator scores. We show that, once organic biases related to spatial and temporal distributions are considered, citizen science data can be used to generate a robust 6.3.2. indicator score. Based on our findings, we are able to make practical recommendations for the integration of citizen science and regulatory monitoring.

## 2. Materials and Methods

### 2.1. Study Locations

We compared conventional regulatory and citizen science (FWW) water quality monitoring schemes in two strategic case study countries: England and Zambia.

Of the 29 countries where Freshwater Watch is active, by far the greatest activity is in the UK, where 10,060 measurements have been made since 2013. Here, FWW is regularly used by 40 volunteer groups who have a specific and often locally-driven interest in regularly monitoring water quality. In addition, Earthwatch Europe have been running biannual 'Waterblitz' events, in which large numbers of volunteers concentrated in various regions coordinate to make as many measurements as possible in a limited 1–4-day event. During these Waterblitz events, volunteers self-select their monitoring sites, and a simplified version of the FWW method is used (turbidity is not recorded). The large number of FWW measurements in the UK make it an ideal case study for exploring the potential of CS data to feed into regulatory monitoring.

Regulatory water quality monitoring is also well established in the UK, which, as a member of the European Union, has been legally obliged to monitor a suite of physical, biological, and chemical indicators of water body 'ecological status' since 2000 as part of the EU Water Framework Directive (WFD) [5]. Implementation of the WFD in the UK is managed by the regulatory authorities of the devolved states of England, Wales, Scotland, and Northern Ireland. Because of the devolved nature of regulatory monitoring in the UK, it is not easy to obtain a single standardized dataset for the whole of the UK. Since the vast majority of FWW data in the UK are located in England, we chose to use regulatory monitoring data from England only.

In Zambia, both regulatory and citizen science monitoring are more limited in their extent. A small group of volunteers, coordinated by the World Wildlife Fund (WWF), in Zambia started using FWW to monitor water quality in the Kafue River in March 2018. There is demand from volunteers to expand FWW in Zambia. The Zambian Water Resources Management Authority (WARMA) are also interested in integrating the data collected by trained volunteers into their national reporting for indicator 6.3.2. Since 2011, as part of the Water Resources Management Act, the Zambian Water Resources Management Authority (WARMA) have been mandated to set ambient water quality standards and, through monitoring and enforcement, to ensure that the standards are maintained [12]. The statutory monitoring network is therefore still in a relatively early stage of development compared to England, and faces challenges related to human and financial capacity for implementation. The expansion of FWW activity from the Kafue River into other areas of Zambia presents an opportunity to design the new citizen science initiatives to compliment the ongoing development of the WARMA statutory monitoring network. This active interest and support from the Zambian NSO warrants its inclusion in this study as it provides a practical application for the analysis.

### 2.2. Description of Datasets

FWW is a global citizen science program run by Earthwatch Europe, and it has been identified as a citizen science project directly relevant to 6.3.2 reporting [3]. Trained volunteers make a number of visual/contextual observations of the water body, including estimates of water depth, water flow, potential pollution sources, and presence of aquatic wildlife, as well as some basic chemical tests for nitrate ($NO_3$-N) and phosphate ($PO_4$-P) and an optical test for turbidity. Chemical tests are conducted in situ using Kyoritsu PackTests (Kyoritsu Chemical-Check Lab, Corp., Tokyo, Japan). Volunteers add a standard 1.5 mL measure of water to a plastic tube containing dry reagents that have an increased absorbance with increasing nutrient concentrations. The resulting color is assigned to one of seven discrete concentration ranges. For nitrates, this colorimetric reaction is produced with the Griess method using zinc powder as the reducing agent [13], and for phosphates an enzymatic reaction using 4-aminoantipyrine with phosphatase is used [14]. Turbidity is measured using a standardized and calibrated turbidity tube [15]. FWW data is available for download through

the website https://freshwaterwatch.thewaterhub.org/. For this study, we downloaded data from 2019 for England and data collected between 2018 and 2020 for Zambia. This captures a full annual cycle of variation in conditions. We also limited the dataset to lotic water bodies (i.e., moving water like streams, channels and rivers), from which the majority of measurements were made in both countries.

Further, 2019 regulatory water quality data for all English water bodies was sourced from the Environment Agency (EA), who are responsible for enacting the EU Water Framework Directive in England. This responsibility includes hydrologically defining 'water bodies' (i.e., 'discrete and significant elements' whose status can be meaningfully quantified [16] (p. 5), setting targets for each water body, and assessing progress towards those targets based on regular monitoring. For this study, data were acquired from the Environment Agency (EA) Water Quality Archive (Beta), available at: https://environment.data.gov.uk/water-quality/view/landing. The data were filtered to exclude compliance data and data from lentic systems, and to include only the determinands nitrate as nitrogen (NO3-N), phosphate as phosphorus (PO4-P), and turbidity (i.e., those variables that are directly comparable to the FWW dataset). Geospatial data defining water body boundaries were obtained from the UK Government's open data portal, accessible here: https://data.gov.uk/dataset/298258ee-c4a0-4505-a3b5-0e6585ecfdb2/wfd-river-waterbody-catchments-cycle-2.

WARMA monitoring data for Zambia is not publicly available for legal reasons. High level geospatial information about the extent of the monitoring network was provided for this research by co-author Nyoni.

For all datasets, we excluded water bodies that were only sampled on one occasion at one monitoring location. We also excluded waterbodies where data on only one of the three parameters were collected. This is in line with the indicator 6.3.2 methodology, and reflects the fact that a single measurement of a water body is not considered to provide a representative understanding of its condition.

### 2.3. Indicator Score Calculation

We generated 6.3.2 indicator scores and statuses for each country as a whole, as well as for the individual water bodies within each country for which data is available. To calculate the indicator scores, we followed an adapted indicator 6.3.2 methodology [17] (see Table 1). This methodology requires that quality status of water bodies is classified using a binary method of either good or not good, using measurements of physico-chemical characteristics. Measurements are compared to numerical targets that represent good ambient water quality. A water body is defined as a discrete hydrological unit such as tributary or section of a river, a lake, or an aquifer. Each individual measurement is compared to its respective target and aggregated to classify the quality at the monitoring station level. Water bodies containing multiple monitoring stations are classified by aggregating monitoring station classifications. Reporting on the indicator is performed on a three-year cycle, and countries are requested to use data from across the preceding three years to the reporting year.

In this study, target values were defined based on the precision of the citizen science methods used. The nutrient test kit methods return discrete rather than continuous values, and therefore targets reflected the boundaries of these concentration ranges. In England, the targets used for both the EA and FWW data were 0.1 mg L$^{-1}$ PO$_4$-P, 5.0 mg L$^{-1}$ NO$_3$-N and 14 NTU for turbidity. These values do not necessarily represent "good ambient water quality" as prescribed in the indicator methodology, but were chosen based on the statistical distribution of the data in order to facilitate the objectives of this study. These target concentrations therefore fall at the boundary between two FWW concentration bands for nutrients and represent the closest category boundary to the median measurement of both datasets combined.

**Table 1.** The application of the official indicator 6.3.2 methodology within this study.

| Indicator 6.3.2 Methodology Component | Application Note |
|---|---|
| In situ water quality measurements of physico-chemical characteristics | Applied |
| Five core ('level 1' parameter groups (nitrogen, phosphorus, salinity, acidity and oxygen), plus optional 'level 2' indicators | Data for two core indicators (nitrogen and phosphorus) and one 'level 2' indicator were used |
| Surface and groundwaters | Focus on lotic waters, for which most data was available. |
| Water bodies defined as discrete hydrological units such as tributaries or inter-confluence sections of river systems | England—the EA's water body units that were defined for WFD reporting [16]. Zambia—HydroBASINS Level 09 units [18] |
| Target value concept | Target values were applied consistently to both EA and FWW data based on thresholds of CS test kits used |
| Data over a three-year period | Data from one year (2019) were selected from both datasets |

## 2.4. Data Analysis

For each country, we compared the proportion of waterbodies monitored by regulatory and FWW schemes to establish the extent to which FWW augmented the existing regulatory networks. For England, where regulatory monitoring and FWW co-existed in numerous waterbodies, we were able to further explore the complementarity between the two datasets by performing Chi-squared tests to compare spatial (stream order) and temporal (sampling period) distributions of data, as well as t-tests to compare the frequency of monitoring (the number of individual data collection events) for each waterbody. This was not possible for Zambia because there were not enough waterbodies with available regulatory and FWW data to make statistically robust comparisons.

We then compared indicator scores and statuses produced by each dataset, both at national level and at water body level. For the latter, we were again able to directly compare the performance of each monitoring method for those waterbodies where regulatory monitoring and FWW co-existed. Correlations between scores produced by each method were quantified using Pearson's Product Moment Correlation Coefficients, and the impacts of the identified spatial, temporal, and monitoring frequency biases on the correlation coefficients were investigated.

## 3. Results

### 3.1. Data Coverage

#### 3.1.1. England

The English regulatory dataset (EA) contained 39,451 nitrate, 40,797 phosphate, and 4984 turbidity records. The English citizen science dataset (FWW) contained 1089 nitrate, 1089 phosphate, and 251 turbidity records. These data allowed for a combined coverage of 62% of the total number of water bodies in England, with the majority covered by the regulatory body (Table 2, Figure 1).

**Table 2.** Summary of water bodies monitored in England in 2019.

| | |
|---|---|
| Total no. of water bodies | 4092 |
| Total monitored | 2523 (62%) |
| Monitored by Environment Agency (EA) | 2492 (61%) |
| Monitored by FreshWater Watch (FWW) | 140 (3%) |
| Monitored by both EA and FWW | 111 (3%) |
| Monitored by FWW only | 31 (1%) |
| Monitored by EA only | 2381 (58%) |

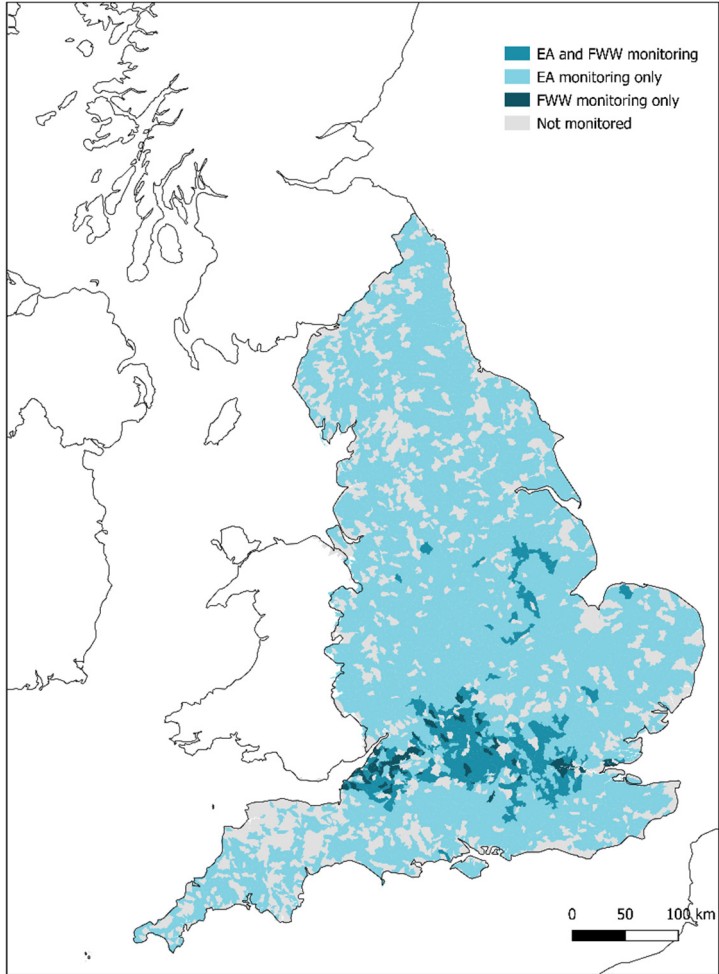

**Figure 1.** Map showing water bodies in England monitored by EA, FWW, both, and neither.

Where both monitoring schemes co-exist, the EA made significantly more measurements per water body than citizen scientists (t = −6.54, df = 108, *p* < 0.0001). However, there were some water bodies where more FWW measurements were taken than EA measurements. Turbidity was the most infrequently measured variable in both monitoring schemes, with only three water bodies for which turbidity data exists from both FWW and the EA. In most water bodies with high numbers of FWW measurements, some or all of these measurements were taken during a Waterblitz event.

### 3.1.2. Zambia

The Zambian regulatory dataset (WARMA) for the Kafue Basin (where FWW activity is regionally contained) has a geographic coverage of 33 water bodies, but the data from these monitoring stations has not yet been ratified for indicator score calculation. The Zambian citizen science dataset (FWW) covered 13 water bodies and contained 68 nitrate, 68 phosphate, and 67 turbidity records. The inclusion of FWW data increased the total number of monitored water bodies in the Kafue Basin by 36%.

### *3.2. Spatial and Temporal Complementarity (England)*

For the English case study, we compared river sizes and the sampling period for all water bodies monitored by both EA and FWW. Water bodies monitored by the EA had an average stream order of 2.66, while that of the English FWW samples was 2.30, which was a significant difference (t = 4.15, *n* = 103, *p* < 0.0001). The resulting distribution of water bodies, considering the average stream order

per water body, was significantly different, and showed an increased selection by citizen scientists of smaller water bodies ($\chi^2$ = 43.6, $p$ < 0.0001) (Figure 2a).

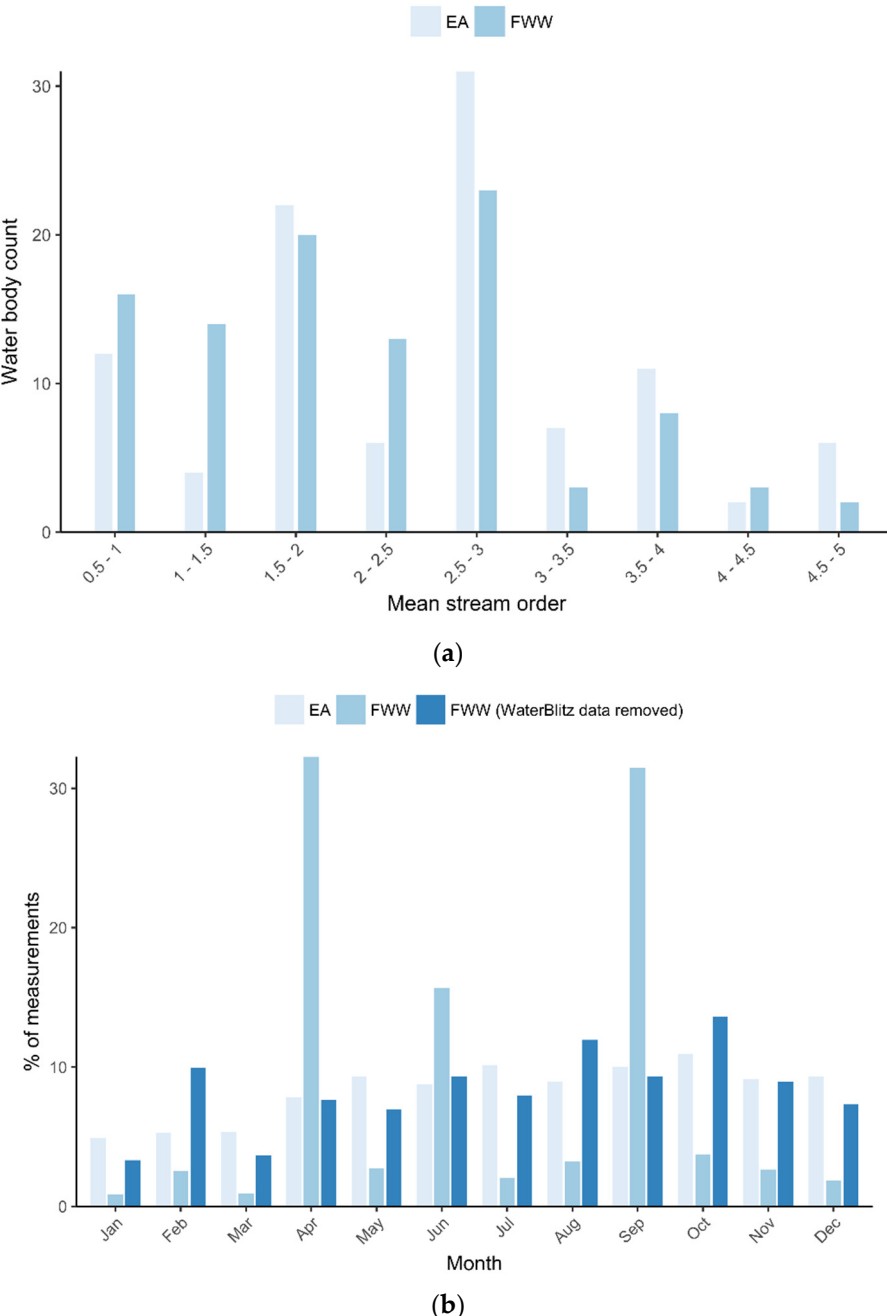

(**a**)

(**b**)

**Figure 2.** Histograms showing (**a**) average stream orders and (**b**) monthly distribution of data for both the EA and FWW datasets.

Measurements across months in each basin (water body) showed clear differences in temporal distribution of the two measurement approaches (Figure 2b) ($\chi^2$ = 1880, $p$ < 0.0001). The frequency of EA measurements showed a more uniform temporal distribution, while FWW measurements showed a strong bias towards the months of April, June, and September, when Waterblitz events were held. Upon removal of Waterblitz data from the FWW dataset, citizen scientist measurements became much more evenly distributed throughout the year, although the distribution of measurements across months remained significantly different from the EA dataset ($\chi^2$ = 25, $p$ = 0.0088).

### 3.3. Indicator Score Calculation

#### 3.3.1. England

The overall indicator score derived from the EA data was 33.51. This was higher than the score derived from the FWW data (22.86) (Table 3). The combined weighted score was dominated by EA data, which covered more water bodies. The relative percentage of water bodies with a score above 80 (pass rate) was also higher for EA data.

**Table 3.** Indicator 6.3.2 scores calculated for England based on EA, FWW, and combined datasets.

|  | No. of Water Bodies Monitored | No. of Water Bodies with 632 Score > 80 (i.e., 'Good' Status) | Overall 632 Indicator Score |
|---|---|---|---|
| EA | 2492 | 835 | 33.51 |
| FWW | 140 | 32 | 22.86 |
| Both datasets combined | 2523 | 833 | 33.02 |

#### 3.3.2. Zambia

Using FWW derived data, 7 out of 10 monitored water bodies were classified 'good', giving an overall indicator score of 70. The locations of these citizen science classified water bodies relative to the existing WARMA network are shown in Figure 3. These represent the only water bodies with ratified and publishable indicator scores at present.

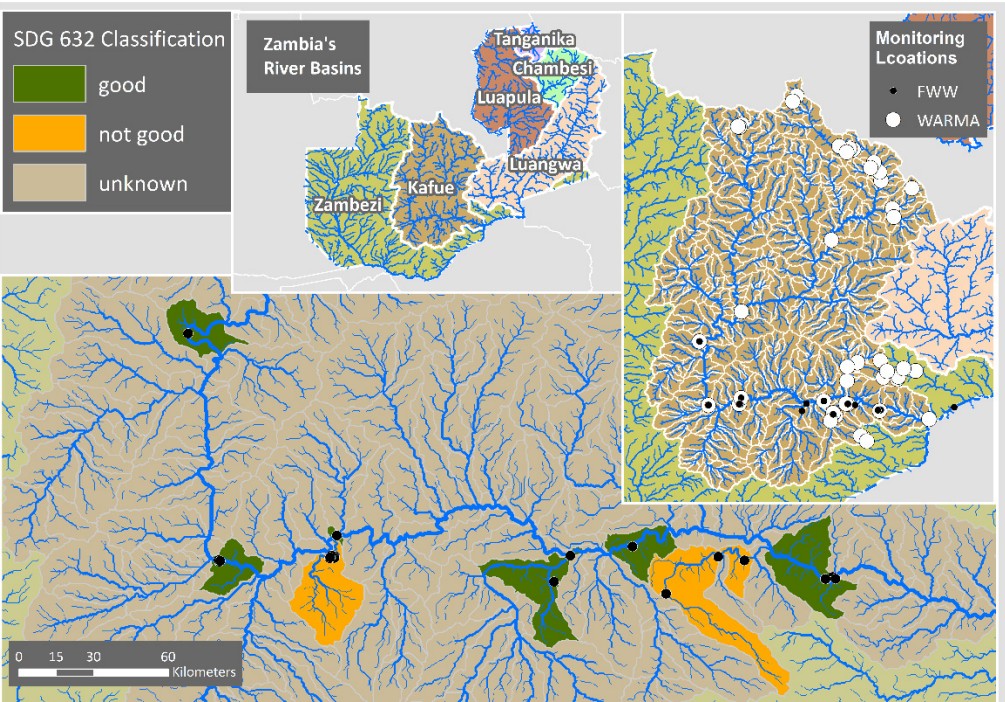

**Figure 3.** Distribution of WARMA monitoring and FWW monitoring in Zambia's Kafue River Basin, showing indicator 6.3.2 classifications calculated based on FWW data (Data source: [18]).

### 3.4. Indicator Score Agreement (England)

For the 111 water bodies that were monitored by both the EA and FWW, we compared indicator scores and statuses produced by both monitoring schemes. There was a low positive correlation between the scores produced by the EA and FWW ($R = 0.37$, $df = 109$; $p < 0.0001$). However, the intercept for the linear model lies at 52.3, suggesting that FWW scores are generally higher than scores produced

using EA data from the same water body, particularly where EA scores are very low (Figure 4). In 22 cases (20%), the schemes produced a different status for the same waterbody.

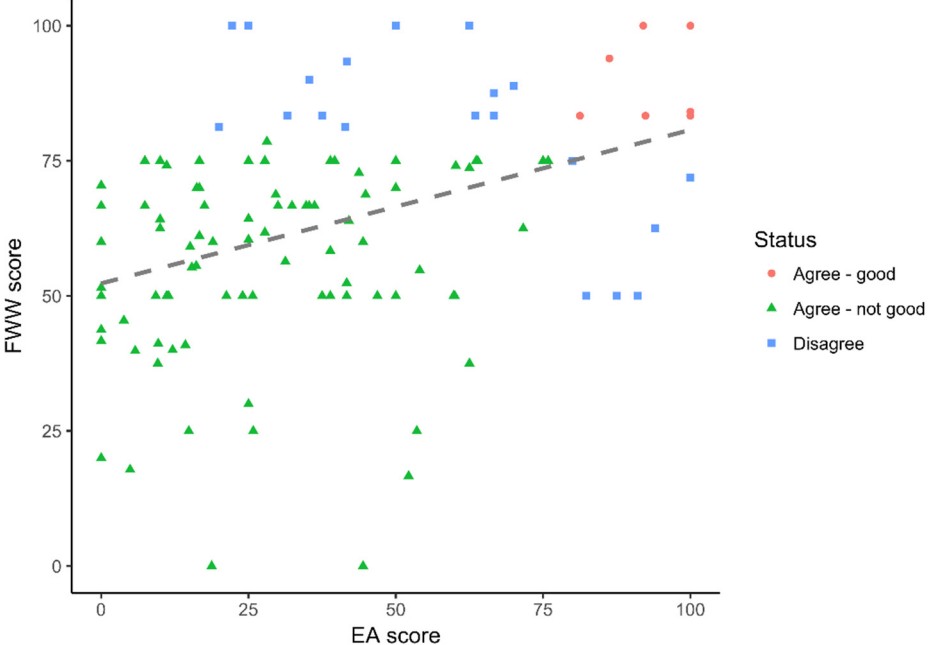

**Figure 4.** Comparison of water body indicator scores generated using EA and FWW data in England. Dotted line shows best fit linear regression line ($R^2$ = 0.13). 'Status' indicates agreement in 632 water body status produced by each method.

The monitoring frequency influenced the strength of the correlation in indicator scores between datasets. Indicator scores for water bodies where fewer FWW measurements were taken had poorer correlations to the indictor scores generated using EA measurements (Figure 5). As the number of FWW measurements increased per water body, the correlation between indicator scores increased. Interestingly, once the number of annual FWW measurements reached 9, there was no significant improvement in correlation with EA measurements with higher frequencies.

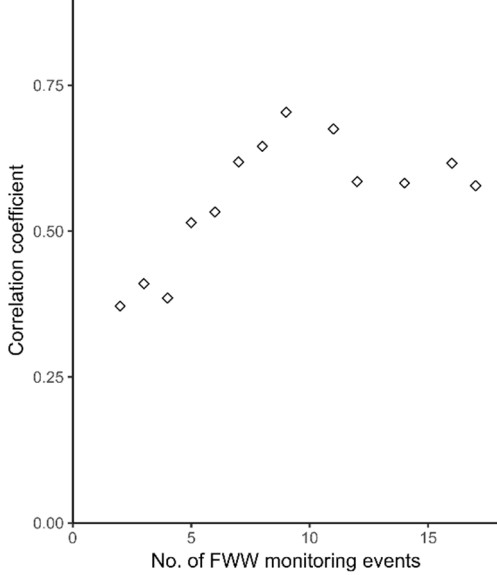

**Figure 5.** Plot showing increasing agreement (Pearson's product moment correlation coefficient) between indicator scores produced using FWW and EA data with increasing FWW monitoring frequency.

## 4. Discussion

A key objective of determining indicator scores for SDG 6.3.2 is to understand and map the conditions of freshwater ecosystems with respect to changes in policy and management of the environment. For this reason, data need to be representative of the water bodies explored and a sufficient number of water bodies needs to be monitored. The results presented in this study highlight the potential to support SDG reporting using citizen science, and provide a basis for recommendations for integrating these approaches into NSO practices in different contexts. Specifically, we highlighted the practical considerations required for integrating data collected via the Freshwater Watch (FWW) citizen science project into NSO data reporting in a scientifically robust manner.

### 4.1. Generating Indicator Scores Using Citizen Science Data

We were able to successfully generate indicator scores for England using both regulatory and FWW datasets, as well as using both datasets combined. For Zambia, it was possible to generate an indicator score based entirely on citizen science data. To do this, two key adaptations were made to the indicator methodology. Firstly, only two of the five core parameters were used. For complete reporting of indicator 6.3.2, measurements of nitrogen, phosphorus, salinity, acidity, and oxygen are required for surface waters, though in practice many member states do not report all five [19]. Although it is possible to measure all five variables using citizen science methods within the FWW setup [10], only nitrates and phosphates are uniformly-used components of existing FWW projects. Although this study demonstrates that it is possible to follow the indicator score calculation mechanisms using existing FWW data, we recommend that future projects incorporate all five core parameters to facilitate complete reporting. Secondly, careful consideration was given to the target values used in order to accommodate the discrete value classification of nutrient concentrations within the FWW method. Because the indicator methodology is designed to be flexible and to accommodate aggregated datasets collected using a variety of methods, these adaptations represent only very minor alterations to the process and should not prevent the acceptance of citizen science data in national reporting.

In England, data collected from the same waterbodies in 2019 demonstrated that the two schemes returned the same indicator score (pass or fail) in 80% of cases. Differences in scores in the remaining 20% of cases were partly explained by differences in sampling location within water bodies, with citizen scientists on average sampling smaller streams. Agreement between regulatory and FWW scores also increased with increasing citizen science sampling frequency. This is likely because the size, seasonality, and temporal variability of water bodies can influence the variation within the data, and the effects of these variations are reduced where multiple measurements are made. Indeed, an oft-quoted advantage of citizen science is that it can be used to increase sampling frequency at relatively low cost [20]. Once these considerations of sampling location and monitoring frequency are accounted for, our results demonstrate that data from the FWW can be used to generate reliable indicator scores for SDG target 6.3.2.

### 4.2. The Value of Integrated Citizen Science/Statutory Monitoring for Indicator 6.3.2

For citizen science to become a valued component of NSO reporting, it is important that citizen science data be considered within the context of existing monitoring programs. Clearly the needs of individual NSOs vary according to data availability. In some countries, particularly in Europe, more or less extensive regulatory water quality monitoring programs already exist for other policy purposes, such as the EU Water Framework Directive [5]. The greatest opportunities for integration of citizen science in these cases lie in establishing and exploiting complementarity between regulatory and citizen science schemes [21].

In the present study, the regulatory monitoring network covers >60% of all water bodies in England, while FWW contributed a relatively limited amount of data. Much of this data was located in water bodies where regulatory monitoring is already well established. However, we found that

citizen scientists do not necessarily measure the same features as conventional monitoring networks, favoring smaller stream orders. In water bodies where regulatory monitoring exists, citizen science therefore complements conventional data collection, providing greater granularity and filling knowledge gaps related to these often under-represented ecosystems [21–23]. Such additional granularity and knowledge of small streams can be particularly important to understand the causes of poor water quality and to pick up early signs of deterioration or improvement.

Moreover, despite the large size of the EA monitoring scheme, there were still 31 water bodies for which the only available data on phosphates and nitrates (core parameters for indicator 6.3.2) were derived from citizen science. Turbidity, which is a level 2 indicator in the 6.3.2 indicator methodology but a core measurement within the Freshwater Watch methodology, is not monitored by the EA as a matter of routine in all water bodies. In this study, 91% of the waterbodies where turbidity data were collected by citizen scientists did not have corresponding turbidity data from the EA (although many did have corresponding data for Total Suspended Solids, which is an equivalent parameter often measured using a digital probe). Outside of FWW and other water quality projects, there are numerous other global citizen science initiatives that focus on recording a variety of level 2 indicators, such as miniSASS which uses benthic macroinvertebrates of streams (www.miniSASS.org) or Lake Observer (https://www.lakeobserver.org/) that incorporates measurements of Secchi depth. The opportunity for using citizen science to gather information from unmonitored waterbodies should therefore not be overlooked, even where extensive national monitoring networks exist.

In countries where regulatory monitoring is under-resourced, citizen science may be the best available source of in situ data for some water bodies. In Zambia, incorporating citizen science data increases the data availability by 36%. In countries with smaller regulatory monitoring operations, citizen science can be an important source of data and potentially the most cost-effective approach to expand monitoring coverage [24]. In such cases, it may be possible for NSOs to partner with citizen science organizations to co-design projects, designed from the outset to provide data for this purpose. This can increase the value of the data for reporting purposes, efficiently targeting key sampling locations and eliminating some of the biases of many existing citizen science projects. Although such schemes can be cost effective, they are not free. Sufficient resources are needed for both project design and long-term support to make such schemes a success.

The decision to incorporate citizen science in SDG monitoring in any NSO context should not be guided only by the value it creates in the form of data. Citizen science also adds value in terms of public engagement and collaboration [11,25]. Involving citizens in water quality measurements engages them with the topic of water pollution, makes them aware of the local situation, and, in doing so, makes it relevant to citizen's lives. This creates support for measures to improve water quality and may also lead to personal behavior change and social network influencing, where participants influence their friends and family to care about the issue too [25]. Thus, citizen science monitoring can additionally contribute to SDG target 6.b: 'Support and strengthen the participation of local communities in improving water and sanitation management.'

*4.3. Recommendations for the Integration of Citizen Science*

4.3.1. Consider Biases within Existing Regulatory and Citizen Science Monitoring Schemes

Regulatory monitoring schemes are often specifically designed to generate a balanced picture across space and time, whereas citizen science schemes may have been created using very different design criteria. For example, when citizen scientists are asked to self-select their sampling locations they tend to prefer areas that they can reach easily, producing a coverage of sites that are more clustered than those made by regulatory agencies [20]. Projects or individual citizen scientists may also focus specifically on sites of concern or local interest rather than the most strategic sites at national scale. Furthermore, our results confirm the tendency for citizen scientists to favor smaller sections of water bodies. Individual citizen science projects will differ in these respects, depending on the citizen science

approach, participant motivations, and purpose of the program. These dataset characteristics should be considered explicitly and can be exploited to provide additional information about a member state's ambient water quality and to inform management action.

### 4.3.2. Design New Citizen Science Monitoring Schemes in Accordance with 6.3.2 Indicator Methodology

Citizen science projects for the purpose of SDG reporting should be designed in line with the guidance provided by UNEP on monitoring program design concepts for SDG indicator 6.3.2 monitoring [19]. UNEP-GEMS/Water recommends monitoring locations as downstream as possible within a water body in order to integrate all upstream influences on water quality. If water bodies are deemed homogenous in terms of water quality then a single monitoring station should suffice. The greater the heterogeneity of a water body the greater the number of monitoring stations are required to ensure that biases from a single location are avoided. This is also true of citizen science data, as shown in this study by the apparent increased agreement between FWW and EA SDG scores with increased citizen science sampling frequency in England. Monitoring frequency should be sufficient to provide data that is representative of the waterbody in question and its seasonal and hydrological patterns. A recommended frequency of once per month or no less than four times per year is suggested for rivers [19].

In Zambia, WARMA are looking to expand regulatory monitoring through the use of citizen science. FWW is currently being used in 13 monitoring locations in the Kafue River basin, and will presently be extended to nine new sites in the Barotse floodplains. For the former, a campaign of comparative measurement sites is providing validation data. For the Barotse, the selection of measurement locations and measurement frequency has yet to be consolidated. Based on the key design parameters in terms of geography and sampling frequency outlined above, we designed an example approach with 20 strategically placed additional sampling locations (Figure 6). At the river basin scale, the potential number of monitoring locations is constrained by both the human and technical resources available. Applying a "top, middle, and bottom" approach to ensure an even distribution across the whole river basin will help avoid a bias towards higher order streams and ensures land use heterogeneity and associated pressures are captured. In the example for the Kafue River basin which encompasses 592 water bodies with a mean area of 264.5 km$^2$ (SD 225.6), monitoring them all was not possible, so a pragmatic approach is needed that optimizes coverage.

In the design phase of any new citizen science activity for indicator 6.3.2 monitoring, such data considerations will need to be combined with the needs of the citizen scientists, e.g., in terms of site accessibility, level of ability, and motivation. Participation dynamics and potential need for additional incentives for continued monitoring will need to be considered to realize the desired data collection. Volunteer fatigue is a common occurrence [26] and previous studies have shown that citizen scientists performing water quality monitoring lose motivation after three to seven years [27].

### 4.3.3. Design and Deliver Citizen Science Monitoring for Multiple Benefits

Citizen science is a broad concept and can be deployed in various ways, with equally varying benefits. Van Noordwijk et al. [25] highlights four citizen science approaches that can be particularly impactful, interest group investigations, captive learning research, place-based community action, and mass environmental census. FWW uses all four approaches, with most of the data used in this study derived through mass environmental census (Waterblitz data) and place-based community action approaches. Place-based projects, setup by local communities who want to monitor and potentially improve their local environment, have strong potential for repeated sampling and provide access to local knowledge that can help understand the local context, enabling solutions. It may therefore be possible to leverage the connection between citizen scientists and their interests to identify and prioritize environmental monitoring and management actions towards individual water bodies [28]. Blitz type events can provide spatially resolute data that allows for a high spatial resolution necessary for the

identification of variations in water quality within a single water body [29]. Furthermore, these events, when well-designed, can capture data from one-off participants in a structured, scientific manner [30]. Given the 'long tail of participation' (i.e., the propensity for the majority of participants to participate only once) that is common within many citizen science projects [9], blitz approaches have the additional benefit of engaging and recruiting new citizen scientists en masse while still providing robust data, potentially leading to greater environmental impact through behavior change and social network influencing [25].

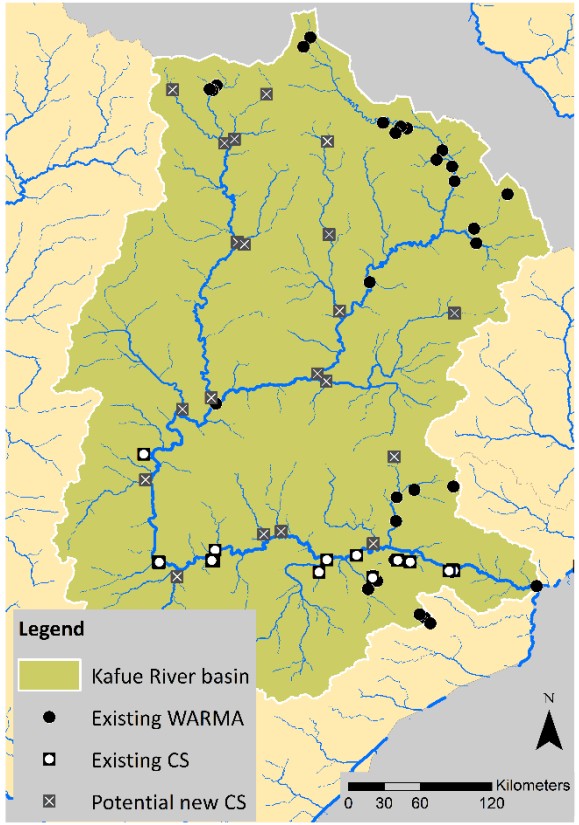

**Figure 6.** Proposed locations for new FWW monitoring in the Kafue River Basin, Zambia (Data source: [18]).

## 5. Conclusions

This study demonstrates that citizen science data can be used reliably for SDG reporting. The citizen science data provide additional insight and granularity, even in countries with extensive statutory monitoring programs. When using data from existing citizen science projects, it is important to be aware of biases in project design and to explicitly use the additional information this generates. The value of citizen science for SDG monitoring is not limited to the value of the data. The public engagement and collaboration generated by the projects can also create environmental impact and help improve water quality. To maximize the value of citizen science for SDG reporting, citizen science projects can be co-designed with statutory agencies specifically for this purpose. In these cases, the design considerations in relation to the data needs should be combined with the needs and practical constraints of the citizen scientists.

**Author Contributions:** Conceptualization, I.J.B., S.W., F.C.N., T.C.G.E.v.N. and S.L.; methodology, I.J.B., S.W. and S.L.; validation, I.J.B., S.W. and S.L.; formal analysis, I.J.B., S.W. and S.L.; investigation, I.J.B. and S.L.; data curation, I.J.B. and S.W.; writing—original draft preparation, I.J.B. S.W. and S.L.; writing—review and editing, I.J.B., S.W., F.C.N., T.C.G.E.v.N. and S.L.; visualization, I.J.B. and S.W.; supervision, S.L. and T.C.G.E.v.N.; funding acquisition, S.L. and T.C.G.E.v.N. All authors have read and agreed to the published version of the manuscript.

**Funding:** The research described in this paper is partly supported by the project COS4CLOUD, which has received funding from the European Union's Horizon 2020 research and innovation program under grant agreement No 863463. The opinions expressed in it are those of the authors and not necessarily those of the COS4CLOUD partners or the European Commission.

**Acknowledgments:** We would like to thank all citizen scientists who contributed to the Freshwater Watch data through Waterblitz events and through various projects in both England and Zambia. We would also like to thank WWF Zambia for coordinating volunteers in the Kafue Basin, Zambia. We are also grateful for feedback from three anonymous reviewers, whose comments led to improvements upon earlier versions of this manuscript.

**Conflicts of Interest:** The authors declare no conflict of interest. The funders had no role in the design of the study; in the collection, analyses, or interpretation of data; in the writing of the manuscript, or in the decision to publish the results.

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
