# Peer review of "Citizen Science Monitoring for Sustainable Development Goal Indicator 6.3.2 in England and Zambia"

_sustainability, doi:10.3390/su122410271_

Round 1

Reviewer 1 Report

In this study, the authors explored the opportunities and biases of citizen science (CS) data when used either as a primary or secondary source for Nations Sustainable Development Goals (SDGs) reporting. They used data from water bodies that have both CS and regulatory monitoring to explore their biases and complementarity. The Case study is England and Zambia
The results showed elevated agreement for pass/fail ratios and indicator scores for English waterbodies (80%). In Zambia, management authorities are actively using citizen science projects to increase spatial and temporal coverage for SDG reporting.

This a very good study. The presentation quality and interest to the readers are high. The authors showed that the results of this study can improve granularity and spatial coverage for SDG indicator scoring and can also address local needs as well as provide a more representative indicator of the state of a nation’s freshwater ecosystems for international reporting requirements.

I did a few corrections into the text (see attached file)

Author Response

Dear Reviewer 1,

Thank you for taking the time to review our manuscript. My co-authors and I are very grateful for your positive, constructive, and encouraging comments, which we have addressed below. Once all of the reviewer’s comments were addressed, the main revisions to the manuscript were some minor edits to the language to improve clarity, and some additional clarification around the need to include all five core parameters in future citizen science work. All edits have been recorded in track changes in the new version of the manuscript.

Specific comments:

Thank you for taking the time to make the edits within the manuscript. We found the following edits, which we agree with and have retained within the latest version:

  1. Removal of spaces between paragraphs
  2. Line 143, 154, 159 – 160, 348, 349 – hyperlinks added.
  3. Re-numbering of section headings in section 4.3

Reviewer 2 Report

Clear, and well done paper. I have no comments.

Author Response

Dear Reviewer 2,

Thank you for taking the time to review our manuscript. My co-authors and I are very grateful for your positive and encouraging comment, which required no specific response from us. Once all of the other reviewer’s comments were addressed, the main revisions to the manuscript were some minor edits to the language to improve clarity, and some additional clarification around the need to include all five core parameters in future citizen science work. All edits have been recorded in track changes in the new version of the manuscript.

Reviewer 3 Report

  1. In Abstract, suggest deleting the word 'more' unless 'more representative' is explained when compared to?
  2. p. 2 line 51. replace 'some countries, particularly in Europe, ..' with 'In Europe. Do not expect authors to be familiar with the U.S. regulatory monitoring program which has been in place since 1972 as required by the Clean Water Act. Presently, the sentence overstates globally.
  3. p 3. line 112. Do not understand why devolution in the UK, makes it difficult to compare among England, Scotland etc. Aren't the protocols for regulatory reporting standardized? Perhaps, a better explanation is that England has more regulatory monitoring sites.
  4. p 4.line 165. 'where data on only one of the three'. 
  5. p 9 line 269. Low positive correlation?
  6. p 10. line 302. Must advocate for using all five core parameters, because using only two to make broad statements on water quality may mislead.
  7. p 11. line 342. The EA may be using Total Suspended Solids (TSS) rather than Turbidity, which is an equivalent measurement.

Author Response

Dear Reviewer 3,

Thank you for taking the time to review our manuscript. My co-authors and I are very grateful for your positive, constructive, and encouraging comments, which we have addressed below. Once all of the reviewer’s comments were addressed, the main revisions to the manuscript were some minor edits to the language to improve clarity, and some additional clarification around the need to include all five core parameters in future citizen science work. All edits have been recorded in track changes in the new version of the manuscript.

Specific reviewer 3 comments:

  1. In Abstract, suggest deleting the word 'more' unless 'more representative' is explained when compared to?

Have deleted the word ‘more’.

  1. p. 2 line 51. replace 'some countries, particularly in Europe, ..' with 'In Europe. Do not expect authors to be familiar with the U.S. regulatory monitoring program which has been in place since 1972 as required by the Clean Water Act. Presently, the sentence overstates globally.

Done.

  1. p 3. line 112. Do not understand why devolution in the UK, makes it difficult to compare among England, Scotland etc. Aren't the protocols for regulatory reporting standardized? Perhaps, a better explanation is that England has more regulatory monitoring sites.

The protocols are standardised to an extent, but some of the published information (e.g. frequency of monitoring, method of monitoring, open-access availability of data) seemed to vary slightly between each nation when we dug into the data. We therefore felt it was better to keep it simple for this study rather than having to standardise the datasets and potentially introducing errors. We have clarified this by changing “it is not easy to compare monitoring data from all states” (line 112) to “it is not easy to obtain a single standardised dataset for the whole of the UK.”  

  1. p 4.line 165. 'where data on only one of the three'. 

Added the word ‘only’.

  1. p 9 line 269. Low positive correlation?

Now reads “there was a low positive correlation…”

  1. p 10. line 302. Must advocate for using all five core parameters, because using only two to make broad statements on water quality may mislead.

We have added the following sentence (line 307): “Although this study demonstrates that it is possible to follow the indicator score calculation mechanisms using existing FWW data, we recommend that future projects incorporate all five core parameters to facilitate complete reporting.”

  1. p 11. line 342. The EA may be using Total Suspended Solids (TSS) rather than Turbidity, which is an equivalent measurement.

We have added the following (line 349): “(although many did have corresponding data for Total Suspended Solids, which is an equivalent parameter often measured using a digital probe)”.

Additional edits following author’s proof-read:

Table 3 column heading edited to read “No. of water bodies with 632 score > 80 (i.e. ‘good’ status)

Line 367 – 368 changed to improve reading. Now reads The decision to incorporate citizen science in SDG monitoring in any NSO context should not  be guided only by the value it creates in the form of data”.